# Nonlinear Optical Properties of Zinc Oxide Nanoparticle Colloids Prepared by Pulsed Laser Ablation in Distilled Water

**DOI:** 10.3390/nano12234220

**Published:** 2022-11-28

**Authors:** Tarek Mohamed, Ali Farhan, Hanan Ahmed, Mohamed Ashour, Samar Mamdouh, Reinhold Schuch

**Affiliations:** 1Laser Institute for Research and Applications LIRA, Beni-Suef University, Beni-Suef 62511, Egypt; 2Department of Engineering, Faculty of Advanced Technology and Multidiscipline, Universitas Airlangga, Surabaya 60115, Indonesia; 3Al Anbar Health Directorate, Ramadi 31001, Iraq; 4High Institute of Optics Technology HIOT, Sheraton Heliopolis, Cairo 11799, Egypt; 5Physics Department, Stockholm University, AlbaNova, 10691 Stockholm, Sweden

**Keywords:** nanoparticles, nonlinear optics, femtosecond laser, high repetition rate, zinc oxide nanoparticles, Z-scan, nonlinear absorption, nonlinear refractive index

## Abstract

The nonlinear optical properties of zinc oxide nanoparticles (ZnONPs) in distilled water were measured using a femtosecond laser and the Z-scan technique. The ZnONPs colloids were created by the ablation of zinc bulk in distilled water with a 532 nm Nd: YAG laser. Transmission electron microscopy, an ultraviolet-visible spectrophotometer, and atomic absorption spectrophotometry were used to determine the size, shape, absorption spectra, and concentration of the ZnONPs colloids. The nonlinear absorption coefficient and nonlinear refractive index were measured at different excitation wavelengths and intensities. The nonlinear absorption coefficient of the ZnONPs colloids was found to be positive, caused by reverse saturable absorption, whereas the nonlinear refractive index was found to be negative due to self-defocusing in the ZnONPs. Both laser parameters, such as excitation wavelength and input intensity, and nanoparticle features, such as concentration and size, were found to influence the nonlinear optical properties of the ZnONPs.

## 1. Introduction

Recently, the development of intense ultrafast lasers has attracted great attention from many researchers for the study of nonlinear optical (NLO) materials [1,2]. Materials with a strong nonlinearity and fast response times are important in many fields, such as nonlinear spectroscopy [3,4], optical switch [5], optical data-storage [6], biomedicine [7,8], and optoelectronic devices [9,10,11]. Future optoelectronic devices are predicted to be small in size with a high performance and high operation speeds. Nanomaterials are thought to be the appropriate medium for these devices [12]. Among nanomaterials, the transparent conducting oxide (TCO) nanomaterial is one of the best candidates for applications in the abovementioned fields. The combination of unique optical and electrical properties of TCO makes it a crucial element in a variety of optoelectronic devices. TCO nanomaterials are wide-bandgap semiconductors (>3.1 eV) characterized by high transparency in the visible range and high carrier density, which makes it electrically conductive [13,14]. TCO nanomaterials are of particular interest in the domain of NLO due to their large optical nonlinearity and short response time [15,16], as well as the fact that their NLO properties can be adapted by controlling the size and shape or doping with metallic ions [12,17].

Zinc oxide nanoparticles (ZnONPs) are one particular type of TCO nanomaterial that has attracted great attention because of their wide bandgap, negligible absorption in the visible region and large exciton-binding energy of 60 meV at room temperature [18], high mechanical and chemical stability, and low cost [19]. This combination of properties has made them a powerful element in many applications, such as electronics [20], nanomedicine [21], bioinspired systems [22], and communications [23]. Owing to their perfect crystalline nature, stability at high temperature and pressure, and high transparency in a wide-band optical region, ZnONPs are considered to be a suitable material in many nonlinear optical devices, including optical limiters, new frequency generation, optical switches, biological imaging and treatment with high resolution [24,25,26], and UV-blue nanolasers [27,28]. Based on the stated applications, there has been a lot of interest in the synthesis and study of the NLO properties of ZnONPs. 

Several physical and chemical approaches, including solgel [29], thermal plasma [30], hydrothermal methods [31], and pulsed laser ablation in liquids (PLAL) [32], have recently been investigated for the liquid-phase synthesis of ZnONPs. PLAL is one of the most important physical approaches for producing NPs from their bulk form. When compared with other methods for obtaining metal colloids, PLAL is advantageous due to its ability to create high-purity nanomaterials using a simpler, faster, and direct methodology that does not require many steps or long process times [33]. It gives a smaller NP size with a narrower size distribution as compared with other conventional techniques [34]. Furthermore, PLAL is a flexible and environmentally friendly approach that enables the synthesis of a wide range of NPs without the use of any toxic or hazardous substances [32,35]. In PLAL, the high-intensity laser light is targeted on the surface of the solid target immersed in liquid. The sizes and shapes of the nanoparticles created by PLAL can be controlled by adjusting the laser beam’s parameters [36]. It is a one-step process that can result in ready-to-use batches without the need for additional sorting procedures [37].

The techniques for studying the NLO properties of ZnONPs are, e.g., nonlinear interferometry, optical Kerr gate, degenerate four-wave mixing, beam-distortion measurements, and Z-scan [38,39]. Z-scan [40] is the most comfortable and informative technique due to its experimental simplicity and high sensitivity compared with the other techniques. Z-scan is also applicable to different materials including liquids [41], semiconductors [42], and glasses [43,44].

Several reports [45,46,47,48,49,50,51] employed the Z-scan technique to explore the NLO properties of ZnO thin films. The results revealed that ZnO thin films had reverse saturable absorption (RSA) and a negative nonlinear refractive index. However, due to the difficulty in creating colloids with high homogeneity and homogeneous sizes and shapes, only a few studies [52,53] have reported on the investigation of ZnO nonlinearity as nanocolloids. Due to thermal effects, the NLO properties of ZnONPs under ns laser pulses are stronger than those under ps laser pulses, according to [52]. On the other hand, [53] looked at the effects of ZnONPs’ shapes and solvents on the NLO properties. ZnO nanospheres have higher NLO properties than ZnO triangular nanostructures. The ZnO nanospheres dispersed in water have higher NLO properties than ZnO nanospheres scattered in 2-propanol. 

In the present work, ZnONPs colloids were prepared in a nanospheres structure by pulsed laser ablation in distilled water at various concentrations by using the second harmonic Nd: YAG laser with a constant fluence of 28.29 J/cm^2^. Furthermore, using femtosecond laser pulses and the Z-scan technique, a comprehensive study of the NLO properties of ZnONPs was carried out at different excitation wavelengths ranging from 740 nm to 820 nm and different excitation intensities ranging from 12.7 to 22.2 GW/cm^2^. The structure of the paper is as follows. The preparation of the ZnONPs colloids and the Z-scan experimental setup are described in Section 2. The experimental results are presented in the third section. The conclusions are presented in section four.

## 2. Experimental Setup

### 2.1. Sample Preparation

Figure 1 shows the experimental setup for preparing the ZnONPs colloids by using the 2nd harmonic Nd: YAG laser system from Spectra-Physics (Quanta-Ray PRO 350), which had a pulse duration of 10 ns and a repetition rate of 10 Hz. A 532 nm laser had a Gaussian beam distribution with a TEM00 with a maximum pulse energy of 1500 mJ per pulse. This pulse energy could be controlled by controlling the discharge voltage of the pump source (flash lamp).

ZnONPs colloids were prepared in distilled water from bulk zinc (rectangular in shape and with a purity > 99%). The laser beam with a constant energy per pulse of 20 mJ was focused by a convex lens with a focal length of 10 cm. By using the knife-edge approach, the focal spot’s size was confirmed to be ∼150 µm. The bulk zinc was placed in a beaker filled with 22 mL distilled water and covered with a glass plate that had a hole that had a similar diameter to the laser beam to reduce the splashes, which were ejected by the ablating beam from the air-water interface. The problem with splashing is the deposition of solution outside the beaker was that it was undesirable because the ejected droplets could reach the focusing lens, resulting in a decrease in irradiation intensity, sample loss, focal-distance changes, and decreased ablation efficiency. The liquid was stirred gently by a magnetic stirrer to homogenize the colloid and remove the particles from the laser path, while the zinc bulk was moved to prevent deterioration during the ablation. The nanoparticles were prepared at different ablation times of 5 and 10 min.

### 2.2. Z-Scan Setup

The setup of the Z-scan technique is shown in Figure 2 [54]. The femtosecond laser pulses were delivered by the INSPIRE HF100 laser system from Spectra-Physics, which was pumped by a mode-locked femtosecond Ti: sapphire MAI TAI HP laser, with ~1.5–2.9 W average power, 80 MHz repetition rate, and wavelengths ranging from 690 to 1040 nm.

The laser beam had a Gaussian distribution with TEM_00_ spatial mode and M_2_ < 1.1. The INSPIRE HF100 laser system operated at the fundamental IR pump wavelengths and two other modes of operation: (i) second-harmonic generator (SHG) and (ii) optical-parametric oscillator (OPO). The SHG mode generated an output in the range of 345–520 nm, achieved by tuning the Ti: sapphire laser and rotating the second harmonic NLO crystal. The OPO mode generated two simultaneous outputs; one of which covered the range 490–750 nm for the signal, and the other 930–2500 nm for the idler for a fixed-pump wavelength of 820 nm, tuned primarily by rotating the OPO crystal. These modes of operation allowed for us to tune the output wavelength from 345 to 2500 nm, which was covered by four exit apertures.

The nonlinear optical properties of ZnONPs colloids were studied by using horizontally polarized pulses with 100 fs FWHM pulse duration and in the wavelength range of 740–820 nm, selected by the fundamental IR pump aperture. The Gaussian laser pulses were tightly focused to the beam waist of 22.4 µm by a lens with 5 cm focal length. The ZnONPs colloids were filled in a 1 mm path-length quartz cuvette, which is less than the Rayleigh length (z_0_). The cuvette was scanned through the focus with a micrometer linear-translation stage. The total transmitted laser power was recorded by a power meter (Newport 843R) for the open aperture (OA) Z-scan in order to determine the nonlinear absorption coefficient. In the closed aperture (CA) Z-scan, which was used to determine the sign and the value of the nonlinear refractive index of the substance, an aperture was placed in front of the power meter and adjusted to transmit a fixed fraction of 30% of the total transmitted light of the OA Z-scan in order to achieve a good balance between sensitivity and signal-to-noise ratio [40]. The experimental error in the obtained NLO parameters was about 10%, which mainly originated from the determination of the irradiance distribution used in the experiment, i.e., beam waist, pulse width, and laser power calibration.

## 3. Results and Discussion 

### 3.1. ZnONPs Colloids Analysis 

The concentration of the prepared ZnONPs colloids was measured by atomic absorption spectrometry (AAS) (Agilent Technologies 200 Series AA). The AAS device worked because each ion or atom absorbed a specific wavelength, and the amount of light absorbed at this wavelength was exactly proportional to the concentration of the substance, which was determined using the Beer–Lambert law. In the AAS, the concentration is normally calculated using a calibration curve obtained with known concentration standards [55]. The concentration of ZnONPs colloids was increased from 5 to 13 mg/L by increasing the ablation time from 5 to 10 min. That also showed that as the ablation time increases, the ablation efficiency improves. The optical transmittance of the ZnONPs colloids was measured in a wavelength range of 200–850 nm by using the UV-Vis–NIR spectrophotometer (Perkin-Elmer, Lambda 950 spectrometer), as shown in Figure 3.

The transmissions of the two prepared ZnONPs samples (5, 13 mg/L) were high in the visible and near-infrared regions; however, they were low in the ultraviolet region. This result agrees with the obtained transmission spectrum in [51]. The energy bandgap was determined from the optical transmission data of ZnONPs colloids by using Tauc’s plot equation [56], which can be written as: (1)$${\left(\alpha h\nu \right)}^{2}=A\left(h\nu -E_{g}\right)$$
where *α* is the linear absorption coefficient, *hν* is the energy of the photons, *A* is an energy-independent constant, and *E_ɡ_* is the bandgap energy. Using Tauc’s equation, the bandgaps of the ZnONPs colloids were found to be 5.28 and 5.10 eV for concentrations of 5 and 13 mg/L, respectively, as shown in Figure 4. 

Figure 5a,b shows the TEM images and size-distribution graph histograms of the colloidal ZnONPs colloids at the different concentrations of 5 and 13 mg/L, respectively. The ablation laser fluence was kept constant at 28.29 J/cm^2^, and the ZnONPs colloids had a spherical morphology. The average sizes of the ZnONPs colloids at concentrations of 5 and 13 mg/L were 19.5, and 16.2 nm, respectively. Due to the photo-fragmentation process being more effective with longer ablation times compared with bulk thermal ablation, the average sample size of 13 mg/L decreased as the ablation time was increased from 5 to 10 min. This observation would be clarified with a description of the physics that goes beyond PLAL’s production of nanoparticles. The PLAL is a straightforward thermal effect that involves heating, melting, vaporizing, and ionizing the target material to produce plasma, from which the matter is expelled [57,58]. However, in the present work, the irradiation of the bulk zinc is more related to the photo-fragmentation of the ablated nanoparticles that accumulate close to the laser spot. Once the amount of created nanoparticles becomes sufficiently enormous, they shield the incident laser’s light. Hence, the incident laser energy reaching the zinc bulk is attenuated, which causes no further increase in the number of nanoparticles. As a result, rather than removing material from the zinc bulk, most of the laser’s energy is used to break up the larger ablated particles into smaller ones [59]. Figure 5b, which depicts this finding, demonstrates how the number of nanoparticles with small sizes increases while the number of nanoparticles with large sizes decreases after a 10-min ablation time. The bandgap of ZnONP colloids decreases from 5.28 to 5.1 eV with decreasing size from 19.5 to 16.2 eV, which disagrees with the theory of electron confinement at the nanoscopic scale [60,61,62]. This theory is only applicable to nanoparticles in the quantum range that are between 1 and 10 nm in size, and is not applicable to particles larger than 10 nm, as shown in Figure 4 and Figure 5.

### 3.2. OA Z-Scan Measurements of ZnONPs 

In the OA Z-scan measurements performed using 100 fs and 80 MHz repetition-rate laser pulses, two samples of ZnONPs colloids with different concentrations were irradiated by an excitation wavelength of 800 nm and different incident peak intensities from 12.7 to 22.2 GW/cm^2^. Figure 6 shows the OA Z-scan traces of the ZnONPs colloids, in which all of the curves show a valley that is induced by reverse-saturable absorption (RSA) behavior (i.e., positive nonlinear absorption coefficient) that agrees with the results in [51,63]. The reason for the RSA behavior may be three-photon absorption (3PA), free carriers’ absorption, excited-state absorption, nonlinear scattering, or a combination of these processes [64]. According to the bandgaps of the two samples, it is reasonable to assume that 3PA occurs, which induces the nonlinear absorption process. By fitting the experimental data of Figure 6a–d using Equation (2) [40], the nonlinear absorption coefficients for the samples were obtained at different incident peak intensities.
(2)$$\Delta T_{OA}=1+\frac{\alpha_{3\;}\;I_{0}^{2}L}{3{\;}^{\left(3/2\right)}{\left[\;1+{\left(\frac{z}{z_{0}}\right)}^{2}\right]}^{2}}$$
where $\Delta T_{OA}$ is the normalized transmittance, $\alpha_{3\;}$ is the 3PA coefficient, *I*_0_ is the peak intensity at the focus (*Z* = 0), and L is the thickness of the sample. The solid curves in Figure 6a–d are the fits using Equation (2).

Figure 7 shows how$\;\alpha_{3\;}$ is affected by the incident laser peak intensity. $\alpha_{3\;}$ increases with an increasing incident laser peak intensity. This is because increasing the laser intensity increases the number of excited electrons in the conduction band (3PA), which thus increases the probability of free-carrier absorption [64]. It is also noticeable that the $\alpha_{3\;}$of 13 mg/L is higher than that of 5 mg/L, reaching 289 × 10^−22^ cm^3^/W^2^ and 497 × 10^−22^ cm^3^/W^2^, respectively. This is because increasing the concentration of ZnONPs colloids from 5 to 13 mg/L increases the number of nanoparticles in the solution, which increases the amount of three-photon absorption [54]. According to the previous report [51], the obtained 3PA coefficient is three times larger than that of thin film, and eight times larger than that of bulk zinc.

With the aim of investigating the dependence of the nonlinear absorption coefficient on the excitation wavelength for the ZnONPs colloids, the OA Z-scan was performed at a constant peak intensity of 15.8 GW/cm^2^ and different excitation wavelengths of 740 nm, 760, 780, 800 nm, and 820 nm, as shown in Figure 8a–e. This wavelength range was selected for the purpose of preventing high linear absorption, as it is far from the absorption peak of ZnONPs colloids. 

The data in Figure 8a–e was fitted by using Equation (2) to extract the nonlinear absorption coefficients at different excitation wavelengths. Figure 9 shows that the 3PA coefficient increases from 123 × 10^−22^ to 280× 10^−22^ cm^3^/W^2^ for the concentration of 5 mg/L and from 241× 10^−22^ to 718× 10^−22^ cm^3^/W^2^ for the concentration of 13 mg/L as the excitation wavelengths increases from 740 nm to 820 nm. This indicates a significant dependence of the ZnONPs colloids nonlinear-absorption coefficient on the excitation wavelength domain spanning from 740 nm to 820 nm.

### 3.3. CA Z-Scan Measurements of ZnONPs

The CA Z-scan of the ZnONPs colloids was performed using 100 fs and 80 MHz high repetition rate (HRR) laser pulses. The HRR causes thermal heating in the sample due to the absorption of a portion of the laser energy and its conversion to heat. As a result, during the time between laser pulses, the sample does not return to its initial equilibrium temperature, thus resulting in heat accumulation. This heat can cause a temperature distribution, which in turn causes a change in the distribution of the refractive index. The change in refractive index causes accumulative thermal heating which alters the divergence and convergence of the next incident pulses, resulting in the incorrect interpretation of the CA Z-scan measurements. The separation time between laser pulses in the experiment was ⋍12.5 ns (80 MHz), which was less than the thermal characteristic time of the sample (t_c_ = w/4D), where w is the diameter of the laser beam and D is the thermal diffusion coefficient (D = k/ρC_p_), which is dependent on the sample parameters (k thermal conductivity, ρ the sample density, and C_p_ the specific heat capacity). The thermal characteristic time estimated for liquids and some optical glasses is higher than 40 µs [65]. To overcome the problem of accumulative thermal lensing in this study, a new method was used, which is described in [66] as: (3)$$\frac{1}{f\left(Z\right)}=\frac{{a\;L\;E}_{p}F_{l}{}^{\frac{3}{2}}}{\omega {\left(Z\right)}^{2}}\left(1-\frac{1}{\sqrt{N_{p}}}\right)$$
where a is the fitting parameter, a = α (dn/dT)/2κ (π^3^D)^1/2^, dn/dT is the temperature derivative of the refractive index, E_p_ is the energy per laser pulse, ω(z) is the radius of the laser beam at the sample, F_l_ is the repetition rate, L is the sample thickness, and N_p_ is the number of laser pulses incident on the sample. N_p_ = t × F_l_, where t represents the exposure time for each ZnONPs sample during the scan. The normalized transmittance of the CA Z-scan measurements ΔT_CA_, the features of which depend on the focal length of the induced lensing f(Z), is given by [67]: (4)$${\Delta T}_{\mathrm{CA}}=1+\frac{2Z}{f\left(Z\right)}$$

Thus, the nonlinear refractive index (n_2_) can be measured by using the following equation [66,67,68]: (5)$$n_{2}=\frac{\lambda \;w_{0}{}^{2}\;\Delta \phi }{2\;P_{peak}\;L\;}$$
where Δϕ is the phase shift, the P*_peak_* is the peak power, and L is the length of the sample (⋍1 mm). The phase shift can be expressed by [66] as: (6)$$\Delta \phi =\frac{z_{0}}{2\;f\left(0\right)}$$
where *f* (0) is the focal length of the induced thermal lens when the sample is placed at the focus (Z = 0). The CA Z-scan studies were performed for the 5 and 13 mg/L samples of ZnONPs colloids; the samples were irradiated by an excitation wavelength of 800 nm and different incident peak intensities ranging from 12.7 to 22.2 GW/cm^2^. The closed aperture was adjusted to transmit a fixed fraction of 30% from the total transmitted light in the OA, and the results are shown in Figure 10a–d. The experimental data was fitted to the expression given by Equation (4). In Figure 10, the transmission exhibits a pre-focal peak, followed by a post-focal valley. This reveals that ZnONPs colloids exhibit a negative nonlinear refractive index that is induced by the self-defocusing property of the ZnONPs colloids, which agrees with the previous reports [51,63].

Figure 11 shows the dependence of the nonlinear refractive index on the incident peak intensity. The main reasons for nonlinear refraction in semiconductor materials are the absorptive processes [69,70]. As a result, increasing nonlinear absorption by increasing the intensity as shown in Figure 7 increases the nonlinear refractive index. The data provides that the nonlinear refractive index increases from 55 × 10^−16^ to 65 × 10^−16^ cm^2^/W for a concentration of 5 mg/L and from 95 × 10^−16^ to 121 × 10^−16^ cm^2^/W for a concentration of 13 mg/L with an increasing the incident peak intensity from 12.7 to 22.2 GW/cm^2^, respectively. The maximum refractive index was found to be at a concentration of 13 mg/L. This is because a higher concentration leads to more ZnONPs, which in turn causes more particles to become excited and enhance the nonlinear response.

The CA Z-scan was performed at constant peak intensity of 15.8 GW/cm^2^ and at different excitation wavelengths of 740 nm, 760, 780, 800 nm, and 820 nm to study the dependance of the nonlinear refractive index on the excitation wavelength for the ZnONPs colloids, as shown in Figure 12a–e. 

Figure 13 shows a linear dependence of the nonlinear refractive index on the excitation wavelength. As the wavelength increases from 740 to 820 nm, the nonlinear refractive index n_2_ increases from 35 × 10^−16^ cm^2^/W to 83 × 10^−16^ cm^2^/W for 5 mg/L, and increases from 71 × 10^−16^ cm^2^/W to 127 × 10^−16^ cm^2^/W for 13 mg/L. This is because the increase in wavelength is accompanied by an increase in nonlinear absorption; as previously stated, nonlinear refraction is caused by absorptive processes.

Table 1 summarizes the effects of different laser parameters on the nonlinear optical properties of ZnONPs. This table provides a comprehensive overview of the behaviors of ZnONPs of various sizes under the influence of different excitation laser sources.

## 4. Conclusions

In the present work, the ZnONPs colloids were prepared at different concentrations using the second harmonic of a Nd:YAG laser source. The ZnONPs colloids samples were prepared at different ablation times, which affected their bandgaps, sizes, and concentrations. The linear optical properties of the ZnONPs colloids were determined using transmission electron microscopy and an ultraviolet-visible spectrophotometer. The nonlinear optical properties of the ZnONPs colloids were investigated using the Z-scan technique by 100 fs laser pulses at an excitation wavelength range from 740 to 820 nm, with a peak intensity that varied from 12.7 to 22.2 GW/cm^2^. The ZnONPs colloid samples exhibited RSA behavior and three-photon absorption (3PA) at various excitation wavelengths, excitation powers, and concentrations. The closed aperture measurements reveal that the ZnONPs colloid samples exhibited self-defocusing behavior and a negative refractive index. With an increasing laser power, excitation wavelength, and ZnONPs colloids concentration, the 3PA coefficient and nonlinear refractive index increased.

## Figures and Tables

**Figure 1 nanomaterials-12-04220-f001:**
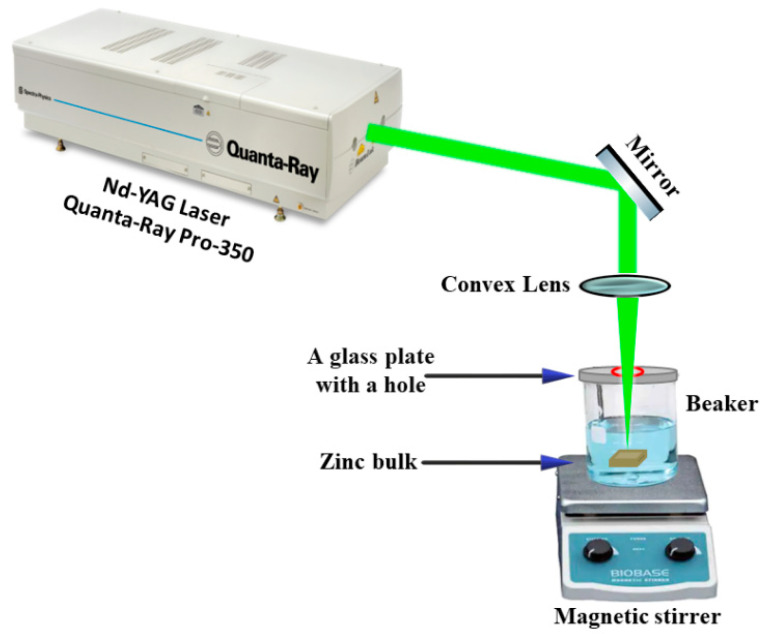
Experimental setup of laser ablation for preparing ZnONPs colloids using 532 nm pulsed Nd: YAG laser.

**Figure 2 nanomaterials-12-04220-f002:**
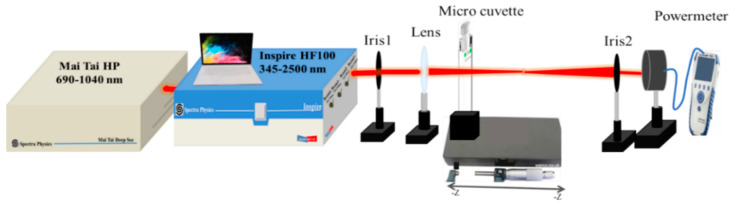
Schematic diagram of Z-scan setup.

**Figure 3 nanomaterials-12-04220-f003:**
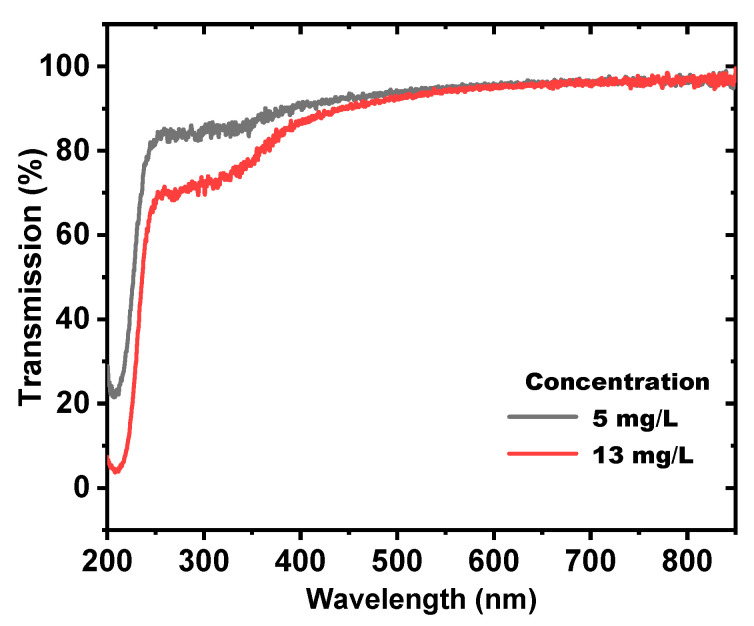
The transmission spectrums of ZnONPs at concentrations of 5 and 13 mɡ/L at 5- and 10-min ablation time.

**Figure 4 nanomaterials-12-04220-f004:**
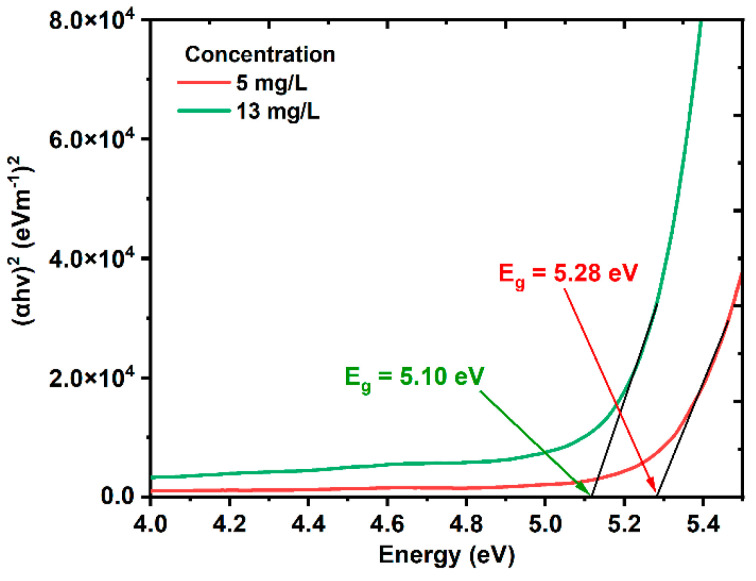
The energy bandgaps of ZnONPs colloids using Tauc’s equation.

**Figure 5 nanomaterials-12-04220-f005:**
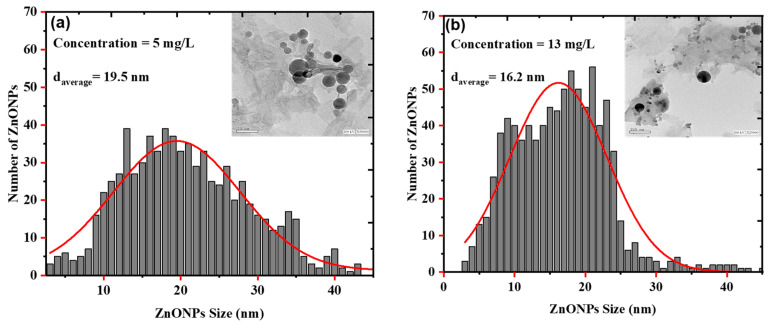
Histogram of the size-distribution of ZnONPs colloids synthesized by laser ablation in distilled water at a constant laser pulse energy of 20 mJ and at different ablation times of (**a**) 5 min (5 mg/L) (**b**) 10 min (13 mg/L), respectively. TEM images of colloidal ZnONPs are included as an inset.

**Figure 6 nanomaterials-12-04220-f006:**
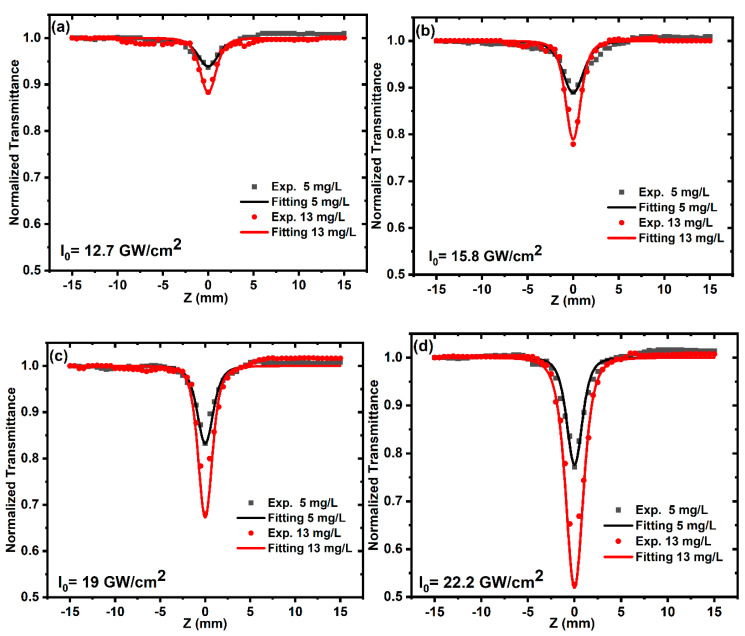
The OA Z–scan transmission of ZnONPs colloids prepared at different concentrations of 5 and 13 mg/L, irradiated by 800 nm, and at different incident peak intensities (**a**) 12.7, (**b**) 15.8, (**c**) 19, and (**d**) 22.2 GW/cm^2^. The points indicate the experimental results, and the solid lines indicate the fits.

**Figure 7 nanomaterials-12-04220-f007:**
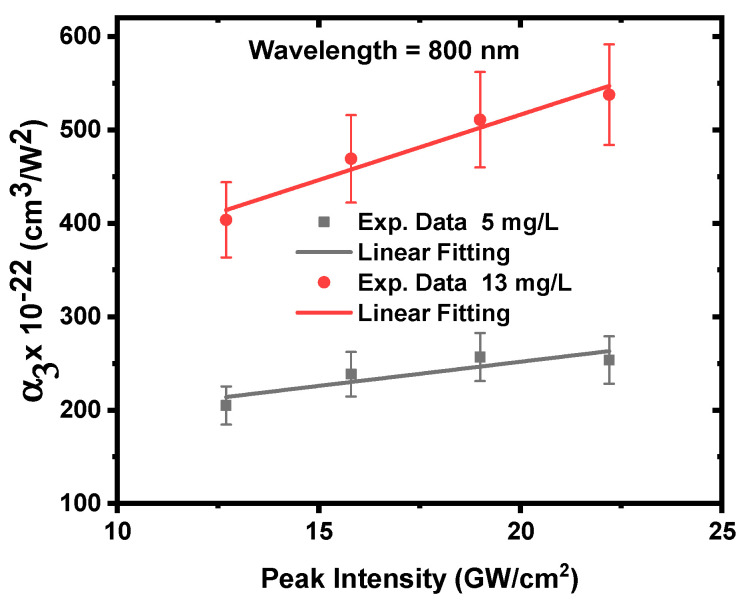
The 3PA coefficient of ZnONPs colloids as a function of the incident peak intensity at a constant excitation wavelength of 800 nm.

**Figure 8 nanomaterials-12-04220-f008:**
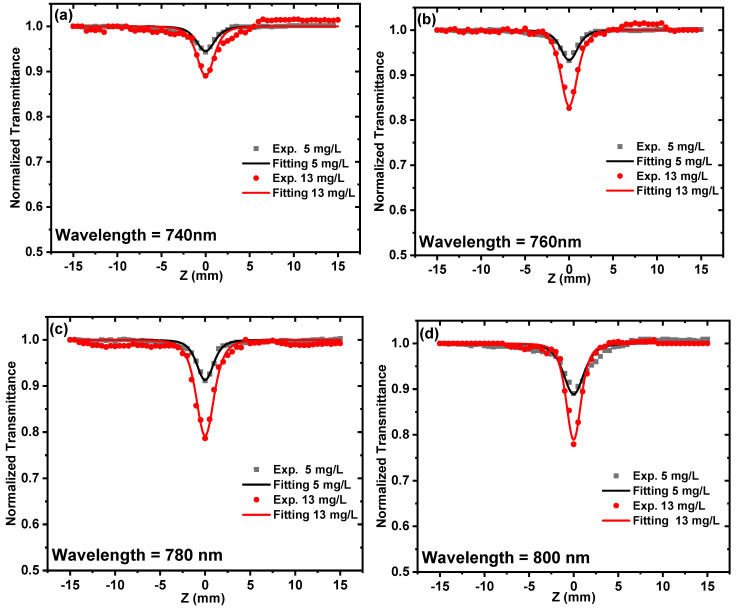
OA Z-scan measurements for the ZnONPs colloids at different excitation wavelengths ranging from 740 to 820 nm at a constant peak intensity of 15.8 GW/cm^2^ (**a**–**e**). The dots indicate the experimental results, and the solid lines indicate the theoretical fitting.

**Figure 9 nanomaterials-12-04220-f009:**
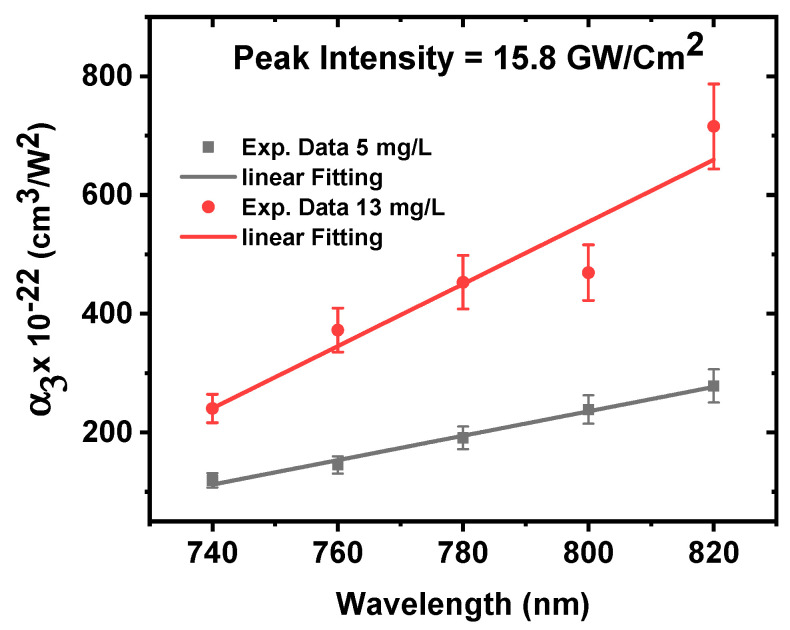
The 3PA coefficient of ZnONPs colloids as a function of excitation wavelength at a constant incident peak intensity of 15.8 GW/cm^2^.

**Figure 10 nanomaterials-12-04220-f010:**
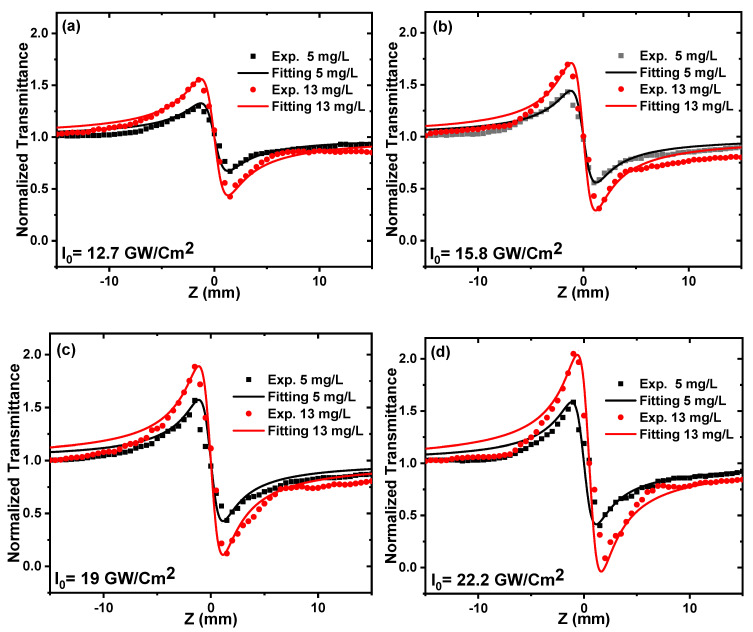
The CA Z–scan transmission of ZnONPs colloids prepared at different concentrations of 5 and 13 mg/L, irradiated by 800 nm, and at different incident peak intensities of (**a**) 12.7, (**b**) 15.8, (**c**) 19, and (**d**) 22.2 GW/cm^2^. The dots indicate the experimental results, and the solid lines indicate the theoretical fitting.

**Figure 11 nanomaterials-12-04220-f011:**
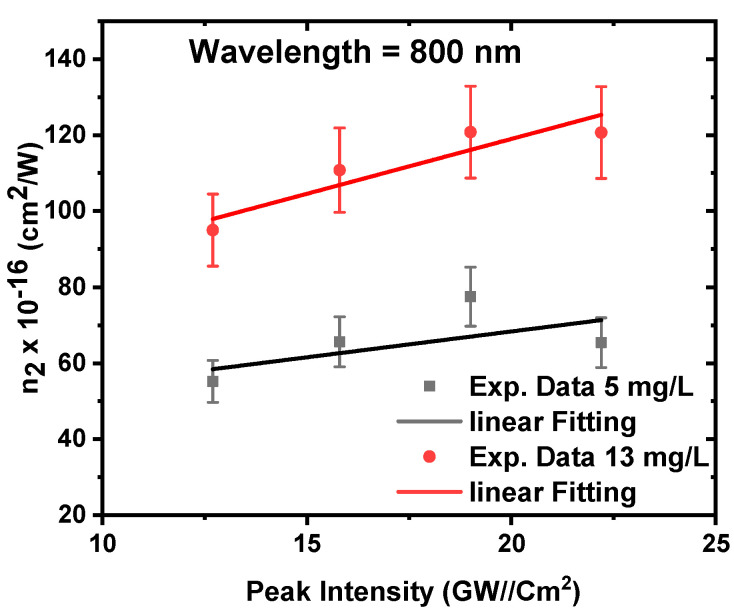
The nonlinear refractive index of ZnONPs colloids as a function of the incident peak intensity at a constant excitation wavelength of 800 nm.

**Figure 12 nanomaterials-12-04220-f012:**
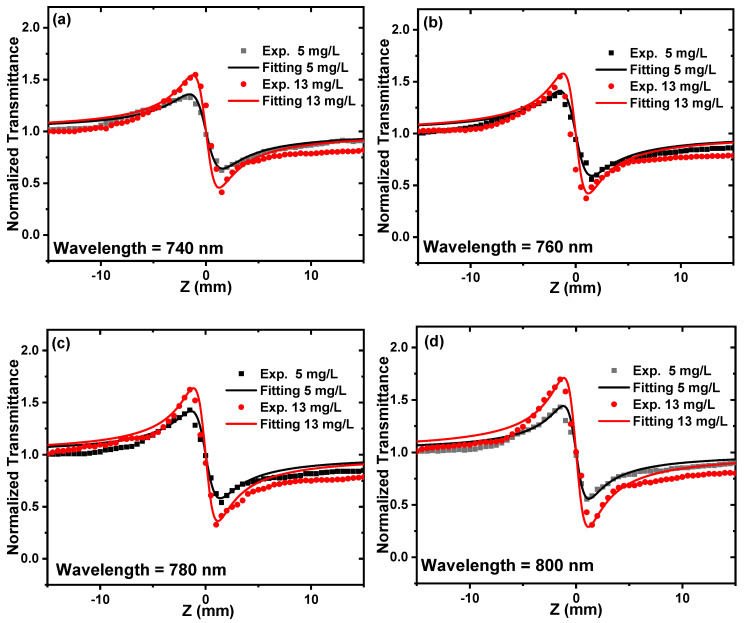
CA Z-scan measurements for the ZnONPs colloids at different excitation wavelengths ranging from 740 to 820 nm and at a constant peak intensity of 15.8 GW/cm^2^ (**a**–**e**). The dots indicate the experimental results, and the solid lines indicate the theoretical fitting.

**Figure 13 nanomaterials-12-04220-f013:**
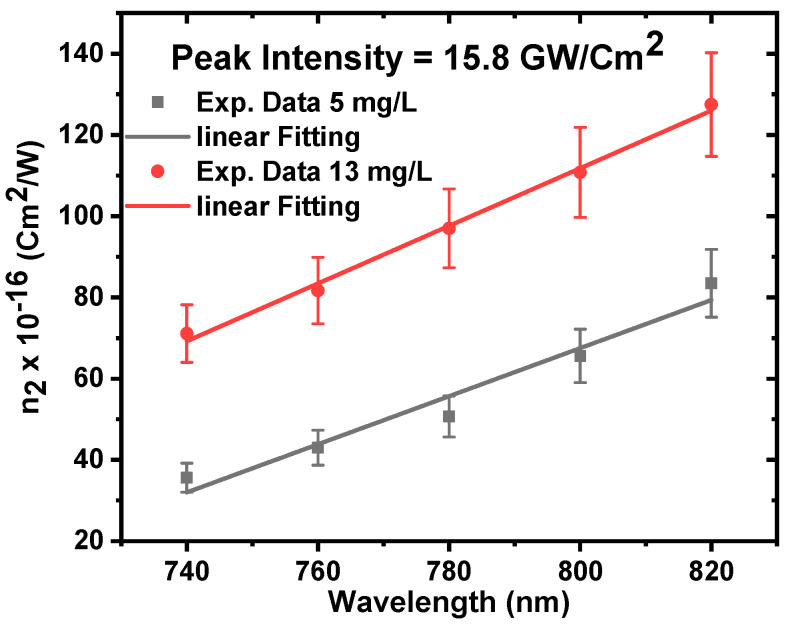
The nonlinear refractive index of ZnONPs colloids as a function of excitation wavelength at a constant incident peak intensity of 15.8 GW/cm^2^.

**Table 1 nanomaterials-12-04220-t001:** Comparison of the nonlinear optical properties of ZnONPs obtained in previous studies and those obtained in this study.

Laser Parameters.			ZnONPs Avg. Size (nm)	*n* _2_ $\left({\mathrm{cm}}^{2}/W\right)$	*β*	Ref.
** λ (nm) **	Pulse Duration	Repetition Rate (Hz)				
532	5 ns	10	100	----	$2.3\times 10^{-11}\mathrm{cm}/W$	[71]
532	10 ns	200	41	$1.2\times 10^{-12}$	$6.6\times 10^{-8}\;\mathrm{cm}/W$	[72]
532	7ns	5	29.5	$1.183\times 10^{-9}$	$5.7\times 10^{-3}\;\mathrm{cm}/W$	[73]
800	170 fs	1000	65	$-1.66\times 10^{-15}$	----	[74]
400 800	40 fs	1000	80 40	$4.3\times 10^{-15}$ $1.6\times 10^{-15}$	$8.4\times 10^{-11}\;\mathrm{cm}/W$ $4.6\times 10^{-11}\mathrm{cm}/W$	[75]
800	100 fs	$80\times 10^{6}$	19.5 16.2	$55\times 10^{-16}$ $65\times 10^{-16}$	$289\times 10^{-22}\;cm^{3}/W^{2}$ $497\times 10^{-22}\;cm^{3}/W^{2}$	Current work

## Data Availability

Data underlying the results presented in this paper are not publicly available at this time but may be obtained from the authors upon reasonable request.

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
