# Peer review of "Nonlinear Optical Properties of Zinc Oxide Nanoparticle Colloids Prepared by Pulsed Laser Ablation in Distilled Water"

_nanomaterials, 2022, doi:10.3390/nano12234220_

Round 1

Reviewer 1 Report

In this manuscript, nonlinear optical properties of zinc oxide 

nanoparticle colloids were measured by Z-scan technique. Such ZnONP 

colloids were fabricated via pulsed laser ablation with a zinc bulk 

in distilled water. The optical properties change with the size and 

concentration of ZnONP colloids. The nonlinear refractive index and 

absorption coefficient also depend on the excitation wavelength 

and intensity. The nonlinear behaviour of ZnONP colloids can be 

attributed to a three-photon absorption process. In addition, the 

authors compared their results with  those in previous work. I 

would like to recommend the manuscript for publication in 

Nanomaterials after several minor revisions. 

1, How to explain that the bandgap of ZnONP colloids increase with 

size, as shown in Figs. 4 and 5?  

2, Is the repetition rate right shown in Table 1? 

3, What will mean if comparing the nonlinear refractive indices and absorption coefficients with previous results?

Author Response

Dear Reviewer,  
We appreciate your excellent remarks on our manuscript.  The detailed revisions are listed below where we present your comments followed by our response.

Point 1: How to explain that the bandgap of ZnONP colloids increase with size, as shown in Figs. 4 and 5? 

Response 1: We would like to thank the reviewer for his/her comments. In accordance with the theory of electron confinement at the nanoscopic scale, as nanoparticle size is reduced, the energy bandgap increases. As discussed in Refs. 1 and 2 in the revised manuscript, this concept is only applicable to nanoparticles in the quantum range that are between 1 and 10 nm in size and is not applicable to particles larger than 10 nm, as shown in Figs. 4 and 5.

Point 2: Is the repetition rate right shown in Table 1? 

Response 2: We appreciate the reviewer's remark. The repetition rate has been checked and corrected in the revised manuscript (80 ×106 Hz).

Point 3: What will mean if comparing the nonlinear refractive indices and absorption coefficients with previous results? 

Response 3: We appreciate the reviewer's comments. It is evident from the results of different reports that the parameters of the prepared nanoparticles, such as size, preparation method, and band gap, as well as the laser parameters, such as wavelength, repetition rate, pulse width, and laser intensity, have an impact on the nonlinear refractive indices and absorption coefficients.

Reviewer 2 Report

The authors use laser ablation technics in water to prepare zinc oxide nanoparticle colloids in different concentrations. They use two methods to determine the distribution of the size and use absorption spectroscopy to find out the concentration. The linear optical properties were determined using transmission electron microscopy and with the help of an ultraviolet-visible spectrometer. They use the z-scan technique with 100fs laser pulses in different wavelengths to find out the non-linear optical properties. These technics are standard technics. They also compare the n_2 and betta values with other authors.

The publication is well written, but some parts have to be improved.

You write to have a confirmed 150 mu laser focus at the zinc oxide sample. How do you measure it?

The fig 7,9,13 needs to have error bars.

Fig 5 has no captions.

I cannot see any central measurement points in fig. 6. Is there a measurement at the curve's minimum? If this is not the case, I don't think it can be a good fit for it.

There might be different aspects playing a role in the production process. The longer the authors use the 532 nm, the higher the possibility that the already produced nanoparticle plays a role in the absorption. If I understand fig 3 right, the transmission of a 5mg/L solution is already only 90%. It would be interesting to publish the transmission curve without nanoparticles in the grave, just for reference.    The production process might be challenging.

Author Response

Dear Reviewer,  
We appreciate your excellent remarks on our manuscript.  The detailed revisions are listed below where we present your comments followed by our response.

Point 1: You write to have a confirmed 150 mu laser focus at the zinc oxide sample. How do you measure it? 

Response 1: We would like to thank the reviewer for his/her comments. The knife-edge approach was used to measure the focus spot's size, which was found to be 150 µm.

Point 2: The fig 7,9,13 needs to have error bars. 

Response 2: We appreciate the reviewer’s remarks, which have already been incorporated into the revised manuscript.

Point 3: Fig 5 has no captions. 

Response 3: We appreciate the reviewer's comments. The caption for Fig. 5 has been checked, and it is already listed in both the original and revised manuscripts.

Point 4: I cannot see any central measurement points in fig. 6. Is there a measurement at the curve's minimum? If this is not the case, I don't think it can be a good fit for it. 

Response 4: We appreciate the reviewer's input. Yes, there is measurable data near the minimum of the curve in Fig. 6. (a- d). We addressed this in the revised manuscript.

Point 5: There might be different aspects playing a role in the production process. The longer the authors use the 532 nm, the higher the possibility that the already produced nanoparticle plays a role in the absorption. If I understand fig 3 right, the transmission of a 5mg/L solution is already only 90%. It would be interesting to publish the transmission curve without nanoparticles in the grave, just for reference.    The production process might be challenging. 

Response 5: We appreciate the reviewer's feedback. We will follow the reviewer's recommendation in our future works with utmost care. The concentration of NPs rises as the ablation time gets longer, and photo-fragmentation of the ablated nanoparticles that accumulate near the laser spot also begins to play a role. The generated nanoparticles screen the incident laser light once their number is sufficiently large. As a result, the incident laser energy is attenuated before it reaches the zinc bulk, which prevents the number of nanoparticles from growing further. As a result, the majority of the laser energy is used to fragment the bigger ablated particles into smaller ones rather than removing material from the bulk of the zinc. Figure 5b, which depicts this finding, demonstrates how the number of nanoparticles with small sizes increases while the number of nanoparticles with large sizes decreases after a 10 minute ablation time.

Reviewer 3 Report

In the manuscript, the authors characterised the nonlinear absorption coefficient and refractive index of zinc oxide nanoparticles dispersed in distilled water at different excitation wavelengths and intensities using femtosecond laser pulses and the Z-scan technique. The nanoparticles were produced by laser ablation, and their size, morphology, absorption spectra, and concentration were determined. The nonlinear optical properties of zinc oxide nanoparticles were found to be dependent on the laser parameters and their size and concentration.

The text is clear, and the covered topic is essential in applications in nonlinear optical devices. The introduction provides sufficient background and includes all relevant references. The authors cite 53 papers in the introduction, including some recent publications from the last five years, and then cite a further 19 papers in the discussion. The manuscript is scientifically sound, and the applied methods are appropriate to the objectives pursued.

In my opinion, the manuscript is acceptable for publication in the journal Nanomaterials after amending the following minor modifications.

1. Line 97. According to the specifications of the Quanta-Ray PRO 350, its max. pulse energy is 2500 mJ at 1064 nm. It should be definitely lower at 532 nm, after the SHG.

2. Fig. 5. While in line 22 20 mJ ablation energy is mentioned in the Methods, in the figure caption it is100 mJ. Please resolve this.

3. The TEM images in Fig. 5. show the colloidal nanoparticles. From the diagrams, the particles appear to be clustered in islands. Can you comment on this? Could this cause the bimodal distribution in figure b?

4. Please define dn in line 292, f(z) f(Z) Z in lines 291, 299.

Author Response

Dear Reviewer,  
We appreciate your excellent remarks on our manuscript.  The detailed revisions are listed below where we present your comments followed by our response.

Point 1: Line 97. According to the specifications of the Quanta-Ray PRO 350, its max. pulse energy is 2500 mJ at 1064 nm. It should be definitely lower at 532 nm, after the SHG. 

Response 1: We appreciate the reviewer's feedback. In the revised manuscript, the maximum energy per pulse for 532 nm has been checked and corrected (1500 mJ/pulse).

Point 2: Fig. 5. While in line 22   20 mJ ablation energy is mentioned in the Methods, in the figure caption it is 100 mJ. Please resolve this. 

Response 2: We appreciate the reviewer’s remarks, which have already been incorporated into the revised manuscript.

Point 3: The TEM images in Fig. 5. show the colloidal nanoparticles. From the diagrams, the particles appear to be clustered in islands. Can you comment on this? Could this cause the bimodal distribution in figure b?. 

Response 3: Thank you for your useful remarks. The tendency of nanoparticles to clump together in solution "agglomeration" is of great interest because the size of the clusters plays an important role in the behavior of the materials. As a follow-up to the work that has been provided, the reviewer's comment is crucial and could be used in our future research keeping in mind that In the current study.

Point 4: Please define dn in line 292, f(z) f(Z) Z in lines 291, 299. 

Response 4: We appreciate the reviewer's comments, which have already been incorporated into the revised manuscript.